# Surgical Management of Chiari Malformation Type I Associated with Syringomyelia: Outcome of Surgeries Based on the New Classification and Study of Cerebrospinal Fluid Dynamics

**DOI:** 10.3390/jcm11154556

**Published:** 2022-08-04

**Authors:** Misao Nishikawa, Toru Yamagata, Kentarou Naito, Noritsugu Kunihiro, Hiroaki Sakamoto, Mistuhiro Hara, Kenji Ohata, Takeo Goto

**Affiliations:** 1Department of Neurosurgery, Moriguchi-Ikuno Memorial Hospital, 6-17-33 Satanakamachi, Moriguchi City 570-0002, Japan; 2Department of Neurosurgery, Graduate School of Medicine, Osaka Metropolitan University, 1-4-3 Asahimachi, Abeno-ku, Osaka 545-8595, Japan; 3Department of Pediatric Neurosurgery, Osaka City General Hospital, 2-13-22 Miyakojimahondori, Miyakojima-ku, Osaka 534-0021, Japan; 4Department of Neurosurgery, Naniwa-Ikuno Hospital, 1-10-3 Daikoku, Naniwa-ku, Osaka 556-0014, Japan

**Keywords:** Chiari malformation, syringomyelia, surgery, clinical outcome, cerebrospinal fluid, cerebrospinal fluid flow dynamics, posterior fossa decompression

## Abstract

Introduction: The mainstay of treatment of syringomyelia associated with Chiari malformation type I (CM-I) is the management of CM-I to normalize the cerebrospinal fluid (CSF) flow at the foramen magnum. CM-I is classified into three independent types. Surgical treatment was selected based on the mechanism of hindbrain ptosis in each CM-I type. Materials and Methods: Foramen magnum decompression (FMD: 213 cases), expansive suboccipital cranioplasty (ESCP: 87 cases), and craniocervical fixation (CCF: 30 cases) were performed. CSF flow dynamics were assessed pre- and post-surgery using cine phase contrast magnetic resonance imaging. During surgery, CSF flow dynamics were examined using color Doppler ultrasonography (CDU). Results: ESCP and FMD demonstrated high rates of improvement in neurological symptoms and signs (82.7%), whereas CCF demonstrated a high rate of improvement in neurological symptoms (89%). The pre-operative maximum flow velocity (cm/s) was significantly lower in patients than in controls and increased post-operatively. During surgery, CDU indicated that the volume of the major cistern was 8 mL, and the maximum flow velocity was >3 mL/s. Conclusions: An appropriate surgical treatment should be selected for CM-I to correct hindbrain ptosis. In addition, it is necessary to confirm the normalization of CSF flow at the foramen of Magendie.

## 1. Background and Introduction

The mainstay of treatment of syringomyelia associated with Chiari malformation type I (CM-I) is the treatment of CM-I to normalize the cerebrospinal fluid (CSF) flow at the foramen magnum. Previously, we reported that CM-I patients have an underdeveloped occipital bone, leading to a shallow posterior cranial fossa (PCF) [1,2,3]. The shallow PCF results in the sagging of brainstem and cerebellum (hindbrain) into the spinal canal; therefore, the pathogenesis of CM-I is insufficiency of a para-axial mesodermal, which is the origin of the occipital bone [1,2,3,4,5,6,7,8,9,10]. Therefore, foramen magnum decompression (FMD) to enlarge the foramen magnum is commonly performed for CM-I worldwide. However, other mechanisms of ptosis of the brainstem and cerebellum, including the hypermobility and instability of the craniocervical junction (CCJ), traction by tethering, and pressure coning [3,11,12,13,14,15], may not respond to FMD. A lack of understanding of the mechanism of ptosis of the brainstem and cerebellum (hindbrain) in CM-I leads to the inappropriate selection of the surgical treatment, which results in the recurrence of neurological symptoms and syringomyelia.

Since 2001, we have performed volumetric and morphometric studies of the PCF using magnetic resonance imaging (MRI) and computed tomography (CT) reconstructed images, which we previously established and evaluated for CM-I patients [1,2,3,11,12]. In addition, we also previously examined the mechanism of ptosis of the brainstem and cerebellum. On the basis of the mechanism of hindbrain ptosis, defined on the basis of volumetric and morphometric analyses, we classified CM-I into three independent subgroups: CM-I types A–C (Table 1; Figure 1) [16]. CM-I type B and CM-borderline are characterized by a normal volume of PCF (VPCF), small volume of the area surrounding the foramen magnum (VAFM), and small occipital bone size (Figure 1B,D). CM-I type C is characterized by a small PCFV, VAFM, and occipital bone size (Figure 1C), as well as the elongation of the brain stem and downward displacement of the whole hindbrain. In CM-I types B and C and CM-borderline, VBPCF/VPCF is large, i.e., the PCF is crowded (Table 1).

In CM-I type A, anatomical abnormalities of the PCF (occipital bone size and hindbrain) were not found, rather only the downward displacement of the hindbrain was observed. CM-I type A (other pathogenesis) is caused by conditions such as craniocervical instability, tethered cord syndrome, hydrocephalus, intracranial mass lesions, and pressure dissociation (Table 1 and Table 2; Figure 1A,E–G) [16]. In this study, we evaluated CCJ hypermobility and instability as another mechanism of downward displacement of the hindbrain.

We selected the surgical procedures for CM-I patients according to the mechanism of hindbrain ptosis (Table 2) [16]. For CM-I type A cases with CCJ hypermobility and instability, craniocervical fixation (CCF) was performed. For CM-I types B and C and CM-borderline, posterior fossa decompression (FMD or ESCP) was performed to correct PCF crowdedness. We present the preliminary results of treatments for the hindbrain ptosis in cases with CM-I associated with syringomyelia. We also examined the CSF flow dynamics at the foramen magnum pre- and post-surgery using cine phase contrast (Cine PC) magnetic resonance imaging (MRI). During surgery, CSF flow dynamics were evaluated using color Doppler ultrasonography (CDU) [17]. We assessed the target value to normalize the CSF flow at the foramen magnum.

## 2. Materials and Methods

### 2.1. Surgical Indication

Surgery was performed in cases of myelopathy, upper cervical cord symptoms, brainstem symptoms, and reliable positive symptoms and neurological findings were determined using the Japanese Orthopaedic Association Cervical Myelopathy Evaluation Questionnaire (JOACMEQ) [18].

### 2.2. Clinical Setting and Patient Selection

#### 2.2.1. Inclusion Criteria

In total, 285 patients with CM-I associated with syringomyelia (cerebellar tonsil herniation ≥ 5 mm from the McRae line, i.e., between the basion and opisthion), brainstem symptoms, and/or myelopathy were included (aged 4–49 years, mean: 18.7 years; 137 males and 148 females), including 14 patients with CM-borderline (cerebellar tonsil herniation < 5 mm from the McRae line, but presence of brainstem symptoms and/or myelopathy). In total, 285 cases were classified into four independent types: CM-I type A (36 cases), CM-I type B (120 cases), CM-I type C (115 cases), and CM-borderline (14 cases).

For the evaluation of CSF spaces in the major cistern and CSF flow dynamics using MRI, 30 normal volunteers (controls) with no neurological symptoms or abnormalities in the neural axis (aged 8–38 years, mean: 17.7 years; 14 males and 16 females) were recruited.

The patients were diagnosed and treated between October 2001 and March 2020. All healthy controls and patients had East Asian ethnicities (Chinese, Korean, or Japanese). The healthy controls and patients were treated at Moriguchi-Ikuno Memorial Hospital, Osaka Metropolitan University Hospital and Osaka City General Hospital.

#### 2.2.2. Exclusion Criteria

We excluded patients with myelopathy and/or radiculopathy due to disc herniation, spondylotic changes, or ossification of the longitudinal ligament. Patients aged older than 50 years were also excluded because of the effect of ageing on the brain and degeneration of the bony structures. Infants and toddlers younger than 4 years were excluded, because of their rapid development and growth, which predisposes to the adverse effects of radiation.

Cases with tethered cord syndrome, hydrocephalus, pressure coning, or pressure dissociation between the intracranial cavity and spinal canal were excluded because they can cause hindbrain ptosis.

#### 2.2.3. Associated Conditions

CM-I and CM-borderline were associated with hereditary disorders of connective tissue in 30 cases (9.3%), basilar invagination in 28 cases (8.7%), other bony anomalies (e.g., assimilation of the atlas, platybasia, os odontoideum) at the CCJ in 57 cases (17.7%).

### 2.3. Assessments and Outcomes

#### 2.3.1. Neurological and Neuroradiological Examination, and Follow up

Post-operatively, the neurological symptoms and signs, JOACMEQ, recovery rate of JOACMEQ score (JOA score RR) ([post-operative points − pre-operative points]/[total points − pre-operative point] × 100%) [18], and neuroradiological findings (cervical spine dynamic X-ray, 2D-CT, and MRI of the cervical spine) were evaluated after every 3 months. The remaining syringomyelia was defined by confirmation of syringomyelia diameter larger than 2 mm.

#### 2.3.2. Volumetric Calculations, Morphometric Measurements, and Data Collection

Volumetric calculations and morphometric measurements were performed by three examiners (except for the main surgeon) who were blinded to the patient identity (M.N, T.Y, K.N, or N.K.). Each examiner repeated the calculations and measurements three times, and the mean data were used.

#### 2.3.3. Statistical Analyses

Surgical outcome was assessed using χ^2^ and Fisher’s tests. A *p*-value < 0.01 was used to determine significance.

### 2.4. Selection of Operative Procedures of Posterior Fossa Decompression and CCF

CM-I was managed using the appropriate surgical methods that can treat the mechanism of hindbrain ptosis [19].

#### 2.4.1. Foramen Magnum Decompression (FMD) for CM-I Type B and CM-Borderline

We performed FMD for CM-I type B and CM-borderline because of VAFM crowdedness due to the small occipital bone size in these cases. The FMD involved craniectomy of the surrounding area of the foramen magnum (2–3 cm square), C1 laminectomy, and dural plasty for decompression of the brainstem and cerebellum, and normalization of CSF flow at the foramen magnum (Figure 2a–d) [20].

In children aged less than 12 years, the cranium and brain are still under development, so the disproportion of the cranium and brain may need correcting. Therefore, in patients younger than 12 years, FMD was performed first. FMD was performed in 213 cases, including 5 cases with CCJ instability, for whom a single-stage CCF was performed.

#### 2.4.2. Expansive Suboccipital Cranioplasty (ECSP) for CM-I Type C

Conversely, in CM-I type C, it is necessary to expand the entire PCF and remodel the relationship between the brain and occipital bone. According to Sakamoto et al. [21], ECSP involves extensive decompression to expand the surrounding area of foramen magnum and the entire PCF (along the transverse and sigmoid sinuses), C1 laminectomy, dural plasty, and osteoplasty to maintain the PCF expansion (Figure 2a–d). ESCP can normalize VPCF and major cisterns, which is adequate for the decompression of the brainstem and cerebellum, and remodel the appropriate positional relationship between the brainstem, cerebellum, and occipital bone by expanding the entire PCF, including C1. We selected ESCP for CM-I type C because FMD was not adequate under the setting of small VAFM, VPCF, and occipital bone size. ESCP was performed in 87 cases, including 4 cases for whom single-stage CCF was performed.

#### 2.4.3. CCF for CM-I with CCJ Instability

CCF was performed for CCJ instability [22,23]. If a patient has instability at the occipito–atlanto–axial joints or occipito–atlantal joints, occipital–cervical posterolateral fixation (OCF) should be performed (Figure 2e). For patients with instability only at the atlanto–axial joints, C1/2 posterolateral fixation (C1/2 FIX) should be performed (Figure 2f). In cases where pedicle screws could be inserted for C2, fixation of C2 with screws is adequate. In cases where screw insertion is not possible, it is necessary to fix C2 to C3 or C4. CCF was performed in a total of 30 cases, of which C1/2 FIX was performed in 16 cases and OCF in 14 cases. For 9 cases with CCJ instability associated with CM-I types B and C, and CM-borderline, staged CCF combined with ESCP or FMD was performed [24].

Therefore, in children (aged less than 12 years), the relationship between anatomical structures was under development; therefore, there is a possibility of overestimating instability. In children, the loss of function of occipital–atlanto–axial joints leads to severe physical disability. Therefore, CCF was not performed for children younger than 12 years.

### 2.5. Evaluation of CSF Flow Dynamics

#### 2.5.1. Evaluation of CSF Flow Dynamics Using Intraoperative CDU

During surgery, CSF flow dynamics were evaluated using CDU (Siemens, Aquarion, Erlangen, Germany) [17]. After bony decompression, before opening the arachnoid membrane and dura mater, and after opening the dura mater and dural plasty, CDU was performed to observe the CSF flow dynamics from the foramina of Magendie and Luschka, and the volume of the major cistern and CSF dynamics were estimated (Figure 3) [17,25].

After the examination was performed, the CSF space was filled with ARTCEREB^®^ solution (Ohtsuka pharmaceutical company, Tokushima, Japan). Milhorat et al. reported that the final CSF space of the major cistern was classified as 8 mL or >8 mL and the maximum flow velocity of CSF as 3 cm/s or >3 cm/s [17]. If the final goal was not achieved, additional procedures were performed (enlargement of the craniectomy, opening of the dura mater and arachnoid membrane, and shrinkage of cerebellar tonsils).

#### 2.5.2. Measurements of CSF Space of the Major Cistern and Evaluation of CSF Flow Dynamics Using MRI Pre- and Post-Operatively

The volume of the major cistern was assessed using MRI (T2 weighted image, axial and sagittal view) (Figure 4). After 3 months, the volume surrounding occipital bone, atlas, cerebellar tonsils, vermis, medulla oblongata, and upper cervical cord was calculated.

Moreover, CSF flow dynamics were assessed pre- and post-operatively using Cine PC MRI. MRI was performed using 1.5 Tesla Philips Gyoscan (Philips, Amsterdam, The Netherlands) (Figure 5). The sequence was synchronized to electrocardiograph gradient echo method, TE 15–20 msec., flip angle 15–30°, and 15 scan/cycle. The cardiac cycle and maximum flow velocity was calculated at the fourth ventricle, outlet of foramen of Magendie, and syringomyelia. The acceleration (cm/ms^2^) of movement of CSF flow was calculated from cardiac cycle and maximum flow velocity (maximum flow velocity [cm/ms.]/% cardiac cycle [ms.]) [26,27,28,29].

## 3. Results

The follow-up duration was 24–234 months (mean: 88.5 months). The results are based on the most recent data. In total, 58 patients were lost to follow up after more than 3 years of operation, in whom the results of the final examination were estimated.

### 3.1. Neurological Symptoms and Signs in Each Type and Its Outcome

The symptoms and signs of the brainstem and cerebellum (headache and neck pain) were significantly more common in CM-I types A and C than in the other groups. The symptoms and signs of spinal cord (pain at extremities, foal numbness, motor weakness, and sensory disturbance) were significantly more common in CM-I type B and CM-borderline than in CM-I types A and C. In CM-I type C, the symptoms and signs (headache, neck pain, ataxia, and dizziness/vertigo) were significantly more common than those of spinal cord myelopathy (pain at extremities, foal numbness, motor weakness, and sensory disturbance) (Table 3).

In CM-I types A–C, the improvement rate of symptoms and signs of brainstem and cerebellum ptosis (headache, neck pain, and ataxia) were higher than those in CM-borderline. In addition, the improvement in pain and sensory disturbance due to spinal cord myelopathy showed lower improvement rates than the brainstem symptoms and signs in all groups (Table 3).

### 3.2. Outcome of Posterior Fossa Decompression (FMD and ESCP)

In patients aged ≥12 years, the improvement rate of neurological symptoms and signs in FMD was 86.5%. The JOA score RR in FMD was 58.7%, while 94.0% of cases showed improvement or stabilization of neurological symptoms. In eight cases (6.2%), the neurological symptoms deteriorated during follow-up. In 10 (7.5%) out of 133 cases with syringomyelia, the symptoms persisted. The improvement rate of neurological symptoms and signs in ESCP was 89.7%. The JOA score RR in ESCP was 60.2%, and 98.9% of cases had an improvement or stabilization of neurological symptoms. In two cases (2.3%), the neurological symptoms deteriorated during follow-up. In 1 (1.1%) out of 87 cases with syringomyelia, the symptoms persisted. There were no significant differences between the FMD and ESCP groups in terms of the improvement or stabilization of neurological symptoms and the JOA score RR. Persistent syringomyelia and deterioration of neurological symptoms and/or signs were significantly more common in patients who underwent FMD compared to ESCP (*p* > 0.01). In eight patients aged 12 years or older with CM-I type B and CM-borderline who had a deterioration of neurological symptoms and signs, ESCP was performed to stabilize the neurological symptoms and signs. In two patients with CM-I type C and deterioration of neurological symptoms and signs after ESCP, CCF was performed, which stabilized the neurological symptoms and signs (Table 4).

In children aged younger than 12 years, the improvement rates of neurological symptoms and/or signs were significantly lower in CM-I types A (58.3%) and C (64.3%) than those in CM-I type B (75.8%) and CM-borderline (71.4%) (*p* > 0.01). Furthermore, JOA score RR was significantly lower in CM-I type C (48.6%) than those in CM-I types A (57.2%) and B (58.2%), and CM-borderline (54.3%) (*p* > 0.01). Persistent syringomyelia and deterioration of neurological symptoms and/or signs were significantly more common in patients with CM-I types A (50.0% and 41.7%, respectively) and C (17.9% and 32.1%, respectively) than those in patients with CM-I type B (6.1% and 7.3%, respectively) and CM-borderline (0% and 14.3%, respectively) (*p* > 0.01) (Table 4).

In six children aged less than 12 years with CM-I type A who had deterioration of neurological symptoms/signs and/or CCJ instability, CCF was performed after 12 years of age, which resolved the neurological symptoms and signs. In 13 children aged less than 12 years and CM-I types B and C and CM-borderline who had deterioration of neurological symptoms/signs, ESCP was performed after 12 years of age, which resolved the neurological symptoms and signs (Table 4).

### 3.3. CCF Outcome

The rate of improvement of neurological symptoms and signs was 90.0%. The JOA score RR among the 30 cases who underwent CCF was 69.7%. In total, 21 cases (70.0%) had complete bony fusion, 25 cases (83.3%) had stabilized joints, and 5 cases (16.7%) had incomplete stabilization. Syringomyelia was resolved in all 21 cases. In all cases, neurological symptoms and signs improved or stabilized (Table 5).

In 16 cases who underwent C1/2 FIX, the rate of improvement of neurological symptoms and signs was 93.8%. The pre-operative JOACMEQ score was significantly higher in C1/2 FIX than in OCF. The JOA score RR was 78.7%. In total, 12 (75.0%), 15 (93.8%), 1 (6.3%) cases had complete bony fusion, stabilized joints, and incomplete stabilization, respectively. In the 14 cases who underwent OCF, the rate of improvement of neurological symptoms and signs was 85.7% and the JOA score RR was 63.5%. In total, 9 cases (64.3%) had complete bony fusion, 10 cases (71.4%) had stabilized joints, and 4 cases (28.6%) had incomplete stabilization. In cases who underwent C1/2 FIX, the JOA score RR, rate of stabilization, and bony fusion of joints were higher than those in patients who underwent OCF (Table 5).

### 3.4. Complications, Side Effects, and Re-Operation Cases

#### 3.4.1. Complication and Side Effects of FMD and ESCP

In the FMD and ESCP groups, there were no cases of mortality or permanent morbidity. Transient morbidity occurred in nine cases (3.0%). Complications were observed in five FMD cases (2.3%) and four ESCP cases (4.6%). Two cases (0.9%) with FMD had wound infection, one case (0.5%) had CSF leakage, one case (0.5%) had arachnoid adhesion, and one case (0.5%) had instability at CCJ. One case (1.1%) with ESCP had wound infection, one (1.1%) had CSF leakage, and two (2.3%) had cerebellar sagging without neurological symptoms. Repetitive and persistent neck pain continued in three patients (1.4%) with FMD and two ESCP patients (2.3%). There were significantly greater complications and side effects in ESCP than in FMD.

#### 3.4.2. Complications and Side Effects of CCF

There was no case of mortality or permanent morbidity after C1/2 FIX or OCF. Transient morbidity occurred in three cases (10.0%). The complications after OCF included transient swallowing disturbance in one case (7.1%) after OCF, wound infection in one case (7.1%), and injury to the vertebral artery without neurological symptoms in one case (6.3%) in C1/2 FIX. Repetitive and persistent neck pain continued in two cases (14.3%) of OCF and functional loss of visual fields occurred in two cases (14.3%). There were significantly greater complications and side effects after OCF than those after C1/2 FIX.

### 3.5. Evaluation of CSF Space and CSF Flow Dynamics

#### 3.5.1. Evaluation by Intraoperative CDU

In ten cases, after bony decompression and dissection of the external layer of dura mater, CSF flow dynamics were evaluated [30]. None of the ten cases achieved normalization of CSF flow (≥8 mL and maximum flow velocity of CSF of ≥3 cm/s). If the goal was achieved, the dissection of the external layer of dura mater was not performed. If the final goal was not achieved, enlargement of bony decompression was performed; however, if the final goal was still not achieved, burning and shrinking of the cerebellar tonsils was performed in 34 cases. Eventually in all the cases, the normalization of CSF flow at the major cistern and CSF flow through the foramina of Magendie and Luschka were achieved.

#### 3.5.2. Pre- and Post-Operative CSF Space in the Major Cistern and CSF Flow Dynamics by Cine PC MRI

The volume of the major cistern was 5.58–10.5 mL (mean: 8.58 mL) in healthy controls, 1.85–7.04 mL (3.56 mL) in CM-I type A pre-operatively, 7.86–12.1 mL (8.78 mL) in CM-I type A post-operatively, 0.25–4.41 mL (3.70 mL) in CM-I type B pre-operatively, 7.55–13.5 mL (8.48 mL) in CM-I type B post-operatively, 0.55–2.42 mL (1.74 mL) in CM-I type C pre-operatively, 9.86–18.5 mL (12.8 mL) in CM-I type C post-operatively, 0.54–5.20 mL (3.77 mL) in CM-borderline pre-operatively, and 7.85–12.3 mL (8.68 mL) in CM-borderline post-operatively. In all types, pre-operative volume was significantly lower than in normal controls. In CM-I types A and B, and CM-borderline, post-operative volume was within the normal range. In particular, in CM-I type C after ESCP, the volume was significantly larger than that in healthy controls (Table 6).

In all types, the pre-operative maximum flow velocity at the outlet of the foramen of Magendie and fourth ventricle were significantly slower than those in healthy controls, with normal post-operative data. In all types, the post-operative maximum flow velocity at the outlet of the foramen of Magendie and fourth ventricle were within the normal range. In syringomyelia, a synchronized rapid flow to the cardiac gate was observed, which disappeared after surgery. In all groups, pre- and post-operative % cardiac cycle was not significantly different compared to the healthy controls. In all types, the pre-operative acceleration of CSF flow from the fourth ventricle through the foramen of Magendie was significantly lower than that in healthy controls, whereas the post-operative values were similar to those of healthy normal controls. In syringomyelia, pre-operative large acceleration was observed; however, post-operatively, reliable data on acceleration could not be calculated.

In cases with deterioration of neurological symptoms and signs, and persistent syringomyelia, the post-operative 6-month maximum flow velocity at the fourth ventricle and outlet of foramen of Magendie were significantly lower than those in healthy controls as well as pre-operative values. Acceleration of CSF flow at the fourth ventricle and outlet of foramen of Magendie were significantly lower than those in healthy controls.

## 4. Discussion

### 4.1. Selection of Surgical Intervention Based on the Mechanism of Hindbrain Ptosis

CM type I should be managed using appropriate surgical methods that can treat hindbrain ptosis [3,16]. Therefore, we selected FMD for CM-I type B and CM-borderline with small VAFM but normal VPCF [19]. Conversely, in CM-I type C, it is necessary to expand the entire PCF and remodel the relationship between the brain and occipital bone. According to Sakamoto et al. [21] ECSP involves extensive decompression to expand the surrounding area of foramen magnum and the entire PCF to maintain PCF expansion. ESCP can normalize VPCF and the major cistern, which is adequate for the decompression of the hindbrain, and to remodel the appropriate positional relationship between the hindbrain, and occipital bone by expanding the entire PCF, including C1. We selected ESCP for CM-I type C because FMD was not adequate under the setting of small VAFM, VPCF, and occipital bone size.

In CM-I type A, other surgical methods that can treat hindbrain ptosis must be selected. CCF should be performed for cases with craniocervical instability causing functional cranial settling [11,13,14,15]. It is important to identify the occipito–atlanto–axial joints with instability using craniocervical traction test, which was previously described by us and Milhorat et al. [11,16,19].

In CM-I types A and C, neurological symptoms and signs of hindbrain are significantly more common than in other types, so it is important to monitor for the development of myelopathy due to syringomyelia, as well as monitor for neurological symptoms and signs due to hindbrain ptosis caused by compression due to narrow VAFM and PCF.

### 4.2. Outcome of Surgical Intervention

#### 4.2.1. Outcomes of Posterior Fossa Decompression (FMD and ESCP)

In cases aged younger than 12 years, in CM-I types A and C, the improvement rate of neurological symptoms and/or signs was significantly lower than that in CM-I type B and CM-borderline. The rates of persistent syringomyelia and deterioration of neurological symptoms and/or signs were significantly higher than those in CM-I type B and CM-borderline. These data suggest that in CM-I type C, FMD was not adequate to decompress the hindbrain or normalize the CSF flow at the foramen magnum; therefore, additional decompression should be considered. In children aged less than 12 years, for cases of CM-I type C, ESCP should be considered or two-staged surgery should be performed, which involves initial FMD followed by observation and ESCP.

In patients aged ≥12 years, ESCP and FMD are associated with good surgical outcomes. The outcomes in the present study were better than those previously reported in cases who underwent FMD only. In addition, morbidity and complications occurred less frequently in the present study than in previous studies [31,32,33]. These findings suggest that the selection of surgical procedures (i.e., ESCP or FMD) according to the morphometric analyses was appropriate. Similar to previous reports, the improvement rate for hindbrain ptosis after FMD and ESCP was high; however, a large proportion of patients with spinal cord myelopathy do not improve after surgery.

In patients younger than 12 years who underwent FMD for CM-I types B and C and CM-borderline, 13 cases had deterioration of neurological symptoms/signs and did not achieve adequate CSF space in the major cistern and normal CSF flow dynamics. Therefore, ESCP was performed after 12 years of age, which resolved their neurological symptoms/signs. In eight patients aged 12 years or older who underwent FMD for CM-I type B and CM-borderline, there was deterioration of neurological symptoms/signs and adequate CSF space at the major cistern and normal CSF flow dynamics were not achieved. Therefore, ESCP was performed, which resolved their neurological symptoms/signs. Therefore, for CM-I type C patients and those who did not achieve adequate CSF space at the major cistern and normal CSF flow dynamics by FMD, the addition of ESCP was appropriate. The addition of ESCP should be considered for CM-I type C patients aged less than 12 years. In 10 preliminary cases with CM-I type C, we initially performed FMD and dural plasty, but could not achieve adequate CSF space volume and CSF maximum velocity; therefore, ESCP was performed. ESCP is a fundamental treatment for achieving adequate decompression of the hindbrain and normalization of CSF flow dynamics in patients with CM-I type C.

We believe that it is essential to intraoperatively confirm the normalization of CSF flow at the foramen magnum. If CSF flow is not normalized with dural opening, more extensive decompression, opening of the arachnoid membrane, lysis of foramina of Magendie and Luschka, and/or shrinkage of tonsils should be considered. Milhorat et al. proposed a tailored operation for patients on the basis of CSF flow dynamics assessed using CDU [17]. We made a decision depending on the need for extensive craniotomy and dural plasty, with or without shrinkage of the tonsils, based on the CSF flow dynamics assessed using intraoperative CDU. The CSF flow may be normalized by decompression with enlargement of craniotomy, second dural opening and plasty, and/or shrinkage of cerebellar tonsils. Knafi et al. previously reported the strategy of salvage and revision surgery [34].

#### 4.2.2. Outcome of CCF for CCJ Instability

For CCF, both C1/2 FIX and OCF achieved high rates of improvement of neurological symptoms and stabilization and/or bony fusion of joints. Our results are superior to those of previous studies of CCF [11,13,14,15,35,36], which suggested that the surgical treatment of CCF was effective for cranial settling due to CCJ instability. OCF leads to a greater loss of function and more frequent complications and side effects due to the fixing of occipito–atlanto–axial joints compared to C1/2 FIX. Therefore, fixation, including that of the occipital bone, should be carefully performed. Craniocervical traction test is useful to detect instability at the occipito–atlantal joints as well as the risk of loss of function due to fixation [11].

The most important factor is to detect instability and hypermobility at the occipito–atlanto–axial joints. Using morphometric analyses during the craniocervical traction test, we were able to detect instability, hypermobility, and dislocation of the occipito–atlanto–axial joints [11]. The morphometric analyses also helped to select the treatment method for joint fixation, determine the symptoms of instability, and confirm whether the instability was reducible. It is necessary to recognize the advantages and disadvantages of joint fixation, including possible loss of joint function. It is necessary to consider the effects on swallowing, neck rotation, and secondary visual loss after OCF and C1/2 FIX.

Therefore, we selected the operative procedure and need for joint fixation on the basis of the results of morphometric analyses performed during the craniocervical traction test. This strategy improved or stabilized the neurological symptoms and/or bony fusion in CCF. In children aged younger than 12 years with CCJ instability and CM-I type A, CCF surgery should be delayed until the patient reaches 12 years of age or the development is complete.

### 4.3. CSF Dynamics and Pathogenesis of Syringomyelia

#### 4.3.1. Study Outcome by MRI and Cine PC MRI

Based on our results, we suggest that the goal of surgery should be based on CSF space and CSF flow dynamics in contrast to improvements of neurological symptom because CSF space and flow are quantitative and objective data. Using posterior fossa decompression (FMD and ESCP) and CCF, normal volumes of the major cistern and normal CSF flow dynamics were achieved and maintained.

#### 4.3.2. Pathophysiology of Syringomyelia Based on our Results

Although % cardiac cycle was similar among patients and healthy controls, the pre-operative maximum flow velocity at the fourth ventricle and outlet of the foramen of Magendie was significantly slower than that in normal controls and returned to the normal range post-operatively. The data were supported by the previous report of Sakas et al. [37]. However, in other reports, Haughton et al. reported that in the patients with syringomyelia, the pre-operative peak systolic CSF velocity was higher than that in healthy controls, and returned to normal post-operatively [26,38]. Previous reports set the region of interest as the whole axial CSF space at the level of the foramen magnum [26,38]. On the other hand, they described that the CSF velocity varied depending on the region of interest [39]. We selected one or two pixels of the outlet of the foramen of Magendie and fourth ventricle as the region of interest. These difference may explain the discrepancy between the results of the present and previous studies. Considering this difference, based on our data, we proposed a hypothesis about the progression of syringomyelia. The pre-operative acceleration was significantly less among patients than in healthy controls, and returned to the normal range in post-operatively. We speculated that pre-operatively, the impulse producing the acceleration of the original CSF movement toward the head promoted the formation of syringomyelia due to the flow of CSF into the spinal cord through the central canal and/or Virchow–Robin space [40]. Therefore, the impulse produced by the acceleration of original CSF flow drove the force for the enlargement of syringomyelia before the surgery. Oldfield et al. and Heiss et al. described that the impulse originated from the piston movement of cerebellar tonsils [41,42]. Chang et al. described a technique of computational fluid mechanics, showing that the major cistern functions as a shock absorber against the pulsatile CSF waves coming from the cranial side. The loss of shock absorbing capacity of the major cistern and subsequent increase in central canal wall leads to syrinx formation [43]. Stoodley and Bliston reported that the impulse originated from the arterial pressure [44]. Although the combination of these three hypotheses can explain the development and progression of syringomyelia, the hypotheses about the pathogenesis of syringomyelia proposed in the present and previous studies alone cannot completely explain the progression and initiation of syringomyelia.

In all previous reports, although hypotheses about the pathophysiology of syringomyelia progression were discussed, the initiation of syringomyelia was never described. There are many unclear aspects of the initiation of syringomyelia, so it is necessary to examine it in greater detail using experimental models (animal models and/or computer system).

## 5. Conclusions

Morphometric analyses of PCF and CCJ should be performed to determine the mechanism and treatment of hindbrain ptosis. CM-I was divided into three independent types based on the morphometric analysis, which suggest the usefulness of morphometric characterization based on the mechanism and pathogenesis of hindbrain ptosis.

The fundamental treatment of syringomyelia associated with CM-I is the management to CM-I and normalization of CSF flow at the foramen magnum. Surgical treatment of CM-I was selected based on the mechanism of hindbrain ptosis in each CM-I type. The surgical treatments had good outcomes and safety profiles. The appropriate surgical method for the treatment of CM-I should address the mechanism underlying hindbrain ptosis. It is important to confirm the normalization of CSF flow during the operation. In the present study, we performed FMD for patients aged younger than 12 years. However, patients younger than 12 years with CM-I types A and C should be monitored for persistent syringomyelia and deterioration of neurological symptoms/signs, which should be treated with ESCP or CCJ after 12 years of age. CCF achieved a high rate of improvement of neurological symptoms and joint stabilization.

We suggest that the goal of surgery would be CSF space and CSF flow dynamics as quantitative and objective data to support normalization of CSF flow at the foramen magnum. Based on our results, we suggest a hypothesis of the progression of syringomyelia.

## Figures and Tables

**Figure 1 jcm-11-04556-f001:**
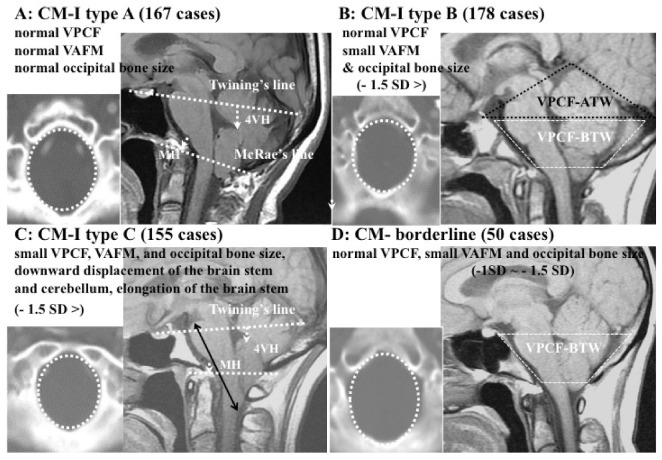
Illustrative cases: CM-I subtypes. (**A**): CM-I type A, (**B**): CM-I type B, (**C**): CM-I type C, and (**D**): CM-borderline. Left panel: 2D CT axial image at the level of the foramen magnum; right panel: magnetic resonance (MR) sagittal midline image. (**E**,**F**): MR midsagittal images demonstrating the character of CM-I type with HDCT. VPCF, VAFM, and occipital bone size are normal in size. (**E**): MR midsagittal image in supine position showing normal interval BDI (7.7 mm), normal BAI (3.5 mm), normal CXA) (141°), large retro-odontoid pannus (asterisk), and low-lying cerebellar tonsils. (**F**): Upon observation of the upright position, there is evidence of cranial settling (2.5 mm decrease in BDI), posterior gliding of occipital condyle, 4.3 mm increase in BAI, anterior flexion of the occipito–atlantal joint (decrease of 8° in CXA), and increased cerebellar ptosis with downward displacement cerebellar tonsils to C1 (white arrow). Note the significantly increased impaction of the foramen magnum anteriorly and posteriorly. Bilateral black dotted arrow = CXA, white arrow = tonsillar herniation. (**G**): MR midsagittal image demonstrating the character of CM-I type A with TCS. VPCF, VAFM, and occipital bone size are normal in size. MR image showing elongation and downward displacement of the brainstem and cerebellum, as well as large supracerebellar cistern (double asterisk). Abbreviations: CM-I = Chiari malformation type I, BDI = the interval between basion and dens, BAI = the interval between basion and atlas, CXA = the clivo-axial angle, HDCT = hereditary disorders of connective tissue, TCS = tethered cord syndrome.

**Figure 2 jcm-11-04556-f002:**
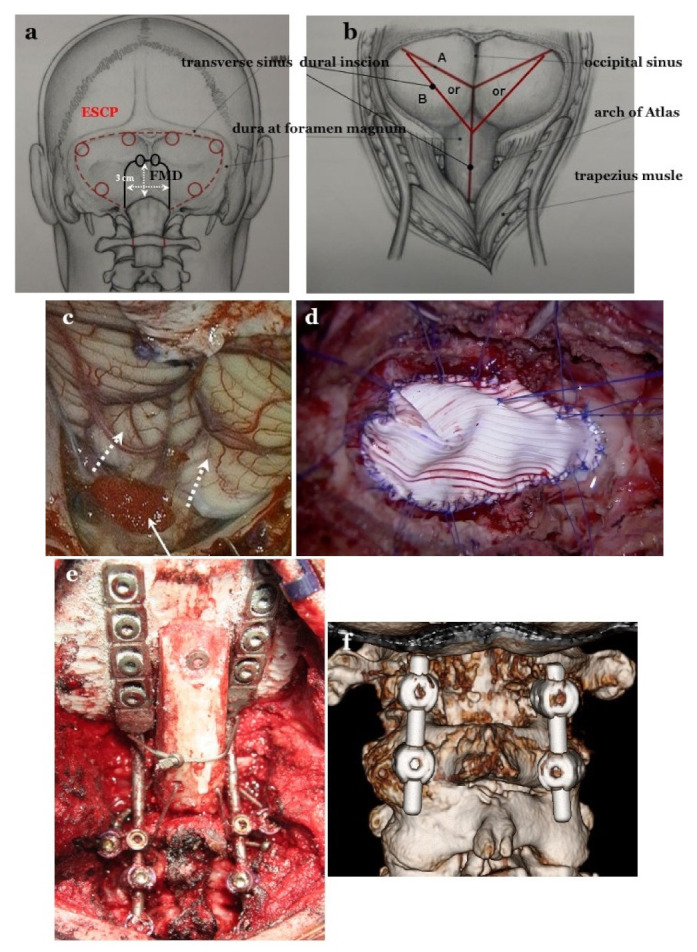
Craniotomy of FMD and ESCP, and dural incision after opening the dura matter and dural plasty. (**a**): For expansive suboccipital cranioplasty (ESCP), craniotomy is performed along the transverse and sigmoid sinuses. The foramen magnum (FMD) craniotomy is 2–3 cm in size. In FMD, the subocciptal muscle group is preserved. (**b**): Dural incision in both operations. (**c**): Opening the dura mater while preserving the arachnoid membrane. Tonsils move up after achieving appropriate decompression (white dotted arrows). Preventing blood from entering into the subarachnoid space through the pinhole by placing a sponge (white arrow) (**d**): Dural plasty by tightly applying Gore Tex^®^ (Gore Inc., Newark, DE, USA) sheet water and applying 6–8 threads on bilateral sutures and midline for tenting. OCF and C1/2 FIX. (**e**): Occipital–cervical fixation (OCF), occipital screws, pedicle screws at bilateral C2, and lateral mass screw at bilateral C3 as anchor screws were connected by rods. (**f**): C1/2 fixation by Goel’s and Harm’s method: pedicle screws at bilateral C2 and lateral mass screws at bilateral C1 as anchor screws were connected by rods.

**Figure 3 jcm-11-04556-f003:**
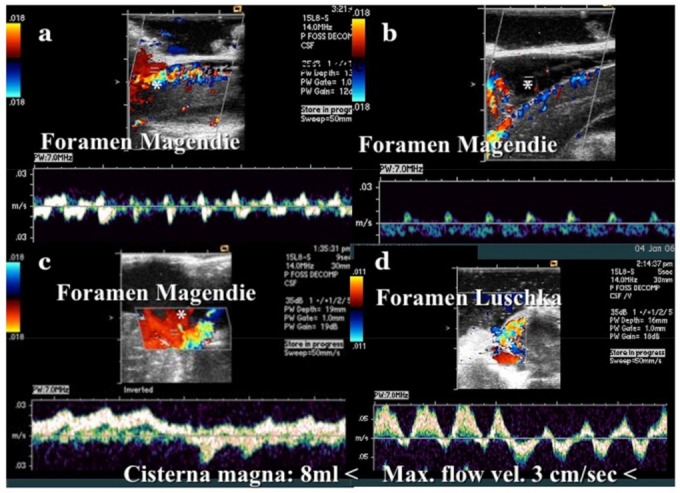
Illustrative cases: color Doppler ultrasonography (CDU). (**a**): CDU at the foramen of Magendie after craniotomy. The cerebrospinal fluid (CSF) space of the cisterna magna was very small. Low CSF stump flow was observed. (**b**): CDU at the foramen of Magendie after opening the dura mater. Although the CSF space became slightly larger than before opening, CSF flow was unchanged, and only low stump flow was observed. * CSF space. (**c**): CDU at the foramen of Magendie after expansive craniotomy and dural plasty. * CSF space. (**d**): CDU at the foramen of Luschka after dural plasty. CDU was performed immediately after craniotomy. In this case, the cisterna magna was small and CSF flow was low, so dural plasty was performed. After dural plasty, CDU was performed again. A sufficiently large wave of CSF flow was confirmed from the foramina of Magendie and Luschka. A sufficiently large space of the cisterna magna (12 mL) was observed post-operatively. The maximum CSF flow velocity was 3 cm/s at the foramen of Magendie and 10 cm/s at the foramen of Luschka.

**Figure 4 jcm-11-04556-f004:**
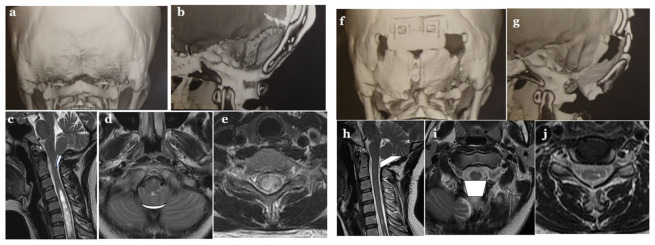
Illustrative cases pre-Op.: (**a**–**e**), Post-Op.: (**f**–**j**).

**Figure 5 jcm-11-04556-f005:**
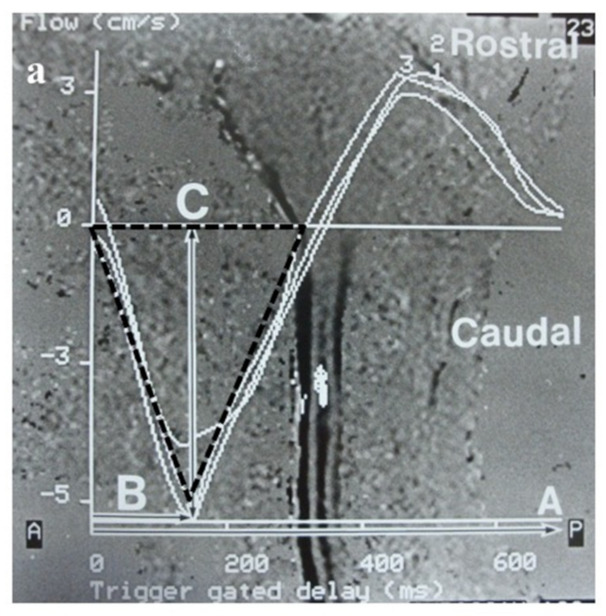
Measurement of maximum flow velocity (cm/s) and % cardiac cycle in Cine PC MRI, and pre- and post-operative illustrative cases. (**a**): Photograph showing the methods used for measuring percentage cardiac cycle (time of maximum flow velocity from R wave: A/R-R interval: B) and maximum flow velocity (speed of maximum flow: C) on cine phase contrast MRI. (**b**–**f**): Cine phase contrast MR images demonstrating that the caudal maximum flow velocity at the foramen of Magendie and dorsal CSF space at the foramen magnum (FM) is smaller than in controls and increased post-operatively. (**d**,**g**): Cine phase contrast MR images demonstrating large caudal maximum flow velocity in the syringomyelia; however, reliable post-operative flow was not confirmed.

**Table 1 jcm-11-04556-t001:** Subtypes of Chiari malformation type I (CM-I) classified on the basis of morphometric analyses, cause of hindbrain ptosis, and surgical intervention.

	CM-I Type A	CM-I Type B	CM-I Type C	CM-Borderline
	36 cases	120 cases	115 cases	14 cases
VPCF	normal	normal	small	normal
VAFM	normal	small	small	small
VBPCF/VPCF	normal	large	large	normal/large
Occipital bone size	normal	small	small	small
Axial length of hindbrain	normal	normal	elongation	normal
Position of hindbrain	downward displacement	normal	downward displacement	normal
Cause	others	crowdedness of AFM	crowdedness of whole PCF	crowdedness of AFM
Intervention	others	FMD	ESCP	FMD

Abbreviations: CM-I: Chiari malformation type I, CM-borderline: Chiari malformation borderline. VPCF: volume of the posterior cranial fossa, VAFM: volume of area of surrounding foramen magnum, VBPCF: volume of brain in the posterior cranial fossa. VBPCF: volume of the area surrounding the foramen magnum, FMD: foramen magnum decompression, ESCP: expansive suboccipital cranioplasty.

**Table 2 jcm-11-04556-t002:** Morphometric characterization and other mechanisms of hindbrain ptosis in CM-I type A, and the corresponding surgical intervention.

	Instability of CCJ	Tethering	Hydrocephalus	Intracranial Mass	Pressure Dissociation
VPCF	normal	normal	normal	normal	normal
VAFM	normal	normal	normal	normal	normal
VBPCV/VPCF	normal	normal	normal/large	normal/large	normal
Brainstem and Cerebellum	normal	elongation and downward displacement		normal	normal
Cause	cranial settling	traction	pressure coning		Hypotension of intraspinal canal
Intervention	CCF	untethering/SFT	VPS	resection of mass or FMD	others

Abbreviations: CM-I: Chiari malformation, CCJ = craniocervical junction, VPCV: volume of the posterior cranial fossa, VAFM: volume of area of surrounding foramen magnum, PFBV: volume of brain in posterior cranial fossa, CCF: craniocervical fixation, SFT: section of filum terminale, VPS: ventriculo-peritoneal shunt.

**Table 3 jcm-11-04556-t003:** Neurological symptoms and signs, and their outcomes.

	CM-I Type A (Craniocervical Instabiity)		CM-I Type B		CM-I Type C		CM-Borderline	
	36 Cases	Improvement Rate	120 Cases	Improvement Rate	115 Cases	Improvement Rate	14 Cases	Improvement Rate
Symptoms and signs of brain stem and cerebellum								
Headache	35 (97.2%) *	33/35 (94.2%) †	76 (63.3%) **	70/76 (92.1%) †	91 (79.1%) *	78/91 (85.7%) †	7 (50.0%) **	16/20 (80.0%)
Neck pain	34 (94.4%) *	32/34 (94.1%) †	74 (61.7%) **	68/74 (91.9%) †	84 (73.0%) *	77/84 (91.7%) †	7 (50.0%) **	13/17 (76.4%)
Ataxia	22 (61.1%) *	14/22 (63.6%)	55 (45.8%) **	45/55 (81.8%) †	75 (65.2%) *	65/ 75 (86.7%) †	5 (35.7%) **	11/15 (73.3%)
Dizziness and/or vertigo	28 (77.8%) *	22/28 (78.5%) †	43 (35.8%) **	33/43 (76.7%)	83 (62.2%) *	73/83 (88.0%) †	2 (14.3%) **	7/10 (70.0%)
Dysphasia	10 (27.8%) **	7/10 (70.0%)	38 (31.7%)	20/38 (52.6%) ⨍	38 (33.0%)	20/38 (52.6%) ⨍	0 (0%)	0 (0%)
Symptoms and signs of spinal cord myelopathy								
Pain of extremities, trunk	16 (44.4%) **	14/16 (87.5%) †	87 (72.5%) *	48/70 (68.6%) ⨍	65 (56.5%)	44/65 (67.7%) ⨍	10 (71.4%) *	8/10 (80.0%)
Focal dysesthesia of extremities and/or trunk	18 (50.0%) **	14/18 (77.8%)	77 (64.2%) *	45/67 (67.1%) ⨍	57 (49.6%)	50/57 (70.2%)	9 (64.3%) *	6/9 (66.7%) ⨍
Motor weakness	24 (66.7%)	12/14 (85.7%) †	60 (50.0%)	47/60 (78.3%) †	51 (44.3%)	45/51 (88.2%) †	11 (78.6%) *	8/11 (72.76%)
Hypalgesia	12 (33.3%) **	6/12 (50.0%)	55 (45.8%)	40/55 (72.7%)	25 (21.7%) **	41/55 (74.5%)	10 (71.4%) *	6/10 (60.0%) ⨍
Dissociated sensory Sensory disturbance	9 (25.0%) **	6/9 (66.7%)	50 (41.7%)	36/50 (72.0%)	20 (17.4%) **	38/50 (76.0%)	8 (57.1%) *	4/8 (50.0%) ⨍

*: significantly more common than those in other groups or other symptoms (*p* < 0.01). **: significantly less common than those in other groups or other symptoms (*p* < 0.001). †: significantly higher than those of other groups or other symptoms (*p* < 0.001). ⨍: significantly lower than those of other groups or other symptoms (*p* < 0.01). Abbreviations: CM-I = Chiari malformation type I, CM-borderline = Chiari malformation borderline.

**Table 4 jcm-11-04556-t004:** JOA score RR after FMD and ESCP, and their outcomes.

	Improved	JOA Score RR (%)	Re-Syringomyelia	Stabilized	Deteriorated	Transient Morbidities
	Neurological Symptoms/Signs	(Mean with ± SD)		Neurological Symptoms/Signs	Neurological Symptoms/Signs	
Total Number: 300 surgeries	248/300 (82.7%)	58.5 ± 10.4	24/300 (8.0%)	25/300 (8.3%)	28/300 (9.3%)	9/300 (3.0%)
(Posterior fossa decompression)						
12 years or older (≥12)						
FMD: 133 cases	115/133 (86.5%)	58.7 ± 10.2	10/133 (7.5%) *	10/133 (7.5%)	8/130 (6.2%) *	3/133 (1.7%)
(for CM-I types A, B and CM-borderline)						
ESCP: 87 cases	78/87 (89.7%)	60.2 ± 10.1	1/87 (1.1%) **	8/87 (9.2%)	2/87 (2.3%) **	4/87 (4.6%) *
(for CM-I type C)						
Younger than 12 years (<12)						
FMD for all types: 80						
CM-I type A: 12	7/12 (58.3%) **	57.2 ± 9.1	6/12 (50.0%) *	0/12 (0%)	5/12 (41.7%) *	1/12 (8.3%)
CM-I type B: 33	25/33 (75.8%)	58.2 ± 10.1	2/33 (6.1%)	5/33 (15.1%)	3/41 (7.3%)	0/33 (0%)
CM-I type C: 28	18/28 (64.3%) **	48.6 ± 10.2 **	5/28 (17.9%) *	1/28 (3.6%)	9 /28 (32.1%) *	1/28 (3.6%)
CM-borderline: 7	5/7 (71.4%)	54.3 ± 10.4	0/7 (0%)	1/7 (14.3%)	1/7 (14.3%)	0 /7 (0%)

*: significantly higher or more than other groups (*p* < 0.01). **: significantly lower or less than other groups (*p* < 0.01). Abbreviations: JOA score RR = the recovery rate of Japanese Orthopaedic Association Cervical Myelopathy Evaluation Questionnaire (JOACMEQ), Re-syringomyelia = syringomyelia persistent, SD = standard deviation, FMD = foramen magnum decompression, CM-I = Chiari malformation type I, CM-borderline = Chiari malformation borderline, ESCP = expansive suboccipital cranioplasty.

**Table 5 jcm-11-04556-t005:** JOA score RR after CCF and its outcome.

Operative Procedures	Improved	Pre-Op. JOACMEQ Score	Post-Op. JOACMQ Score	JOA Score RR (%)	Bony Fusion of Joints	Stabilization of Joints
	Neurological Symptoms/Signs	(Mean with ± One SD)	(Mean with ± One SD)			
CCF: 30 cases	27/30 (90.0%)	4–15 (9.7 ± 2.48)	10–17 (15.5 ± 2.51)	69.7%	21/30 (70.0%)	25/30 (83.3%)
C1/2 FIX 16 cases	15/16 (93.8%)	4–12 (9.0 ± 2.55) *	14–17 (15.4 ± 2.58)	78.7% *	12/16 (75.0%) *	15/16 (93.8%) *
OCF 14 cases	12/14 (85.7%)	3–12 (6.4 ± 2.43) **	10–17 (15.7 ± 2.48)	63.5% **	9/14 (64.3%) **	10/14 (71.4%) **

*: significantly higher than OCF (*p* < 0.01). **: signify lower than C1/2 FIX (*p* < 0.01). Abbreviations: JOACMQ = the recovery rate of Japanese Orthopaedic Association Cervical Questionare, JOA score RR= recovery rate of JOACMQ, SD = standard deviatein, CCF = craniocervocal fixateon, C1/2 FIX = atlanto-axial posterior lateral fixateon, OCF = occipito-cervical fixateon.

**Table 6 jcm-11-04556-t006:** Results of analyses of CSF flow dynamics using Cine PC MRI.

	Controls		CM-I Type B	CM-I Type C	CM-Borderline
Maximum flow velocity (cm/s)					
4th ventricle:	0.68–2.45 (1.96)	Pre-Op	0.86–1.72 (1.48) *	0.55–1.72 (1.17) *	0.56–1.47 (1.28) *
		Post-Op	0.43–2.68 (2.44)	0.41–3.72 (2.18)	0.40–3.78 (2.42)
	Deteriorated cases: post-op 6 months		0.36–0.83 (0.82) **	0.45–1.02 (0.85) **	0.56–0.97 (0.78) **
Outlet of foramen of Magendie	1.70–2.65 (2.24)	Pre-Op	0.45–2.50 (1.32) *	0.38–2.47 (1.42) *	0.44–2.40 (1.44) *
		Post-Op	0.95–5.50 (3.75)	0.84–5.70 (4.54)	0.95 –5.50 (4.25)
	Deteriorated cases: post-op 6 months		0.45–2.50 (0.82) **	0.38–2.47 (0.72) **	0.44–2.40 (0.64) **
Syrinx (spinal cord):	1.25–4.50 (1.98)	Pre-Op	2.75–6.78 (5.35)	2.84–6.72 (4.35)	2.80–7.80 (4.42)
		Post-Op	N.A.	N.A.	N.A.
	Deteriorated cases: post-op 6 months		2.85–7.82 (5.35)	243–5.53 (4.35)	2.67–7.22 (3.25)
% Cardiac cycle (%)					
4th ventricle	35–48 (32)	Pre-Op	28–50 (32)	20–48 (31)	23–51 (33)
		Post-Op	12–52 (35)	9–48 (32)	18–49 (34)
	Deteriorated cases: post-op 6 months		26–52 (34)	22–50 (31)	24–53 343)
Outlet of foramen of Magendie	21–38 (32)	Pre-Op	28–52 (34)	25–50 (36)	24–49 (35)
		Post-Op	14–52 (34)	29–49 (30)	28–47 (34)
	Deteriorated cases: post-op 6 months		28–52 (34)	25–50 (36)	24–49 (35)
Syrinx (spinal cord)	25–45 (38)	Pre-Op	27–47 (35)	25–48 (35)	23–49 (35)
		Post-Op	N.A.	N.A.	N.A.
	Deteriorated cases: post-op 6 months		14–47 (35)	29–49 (33)	29–46 (35)
Caudal acceleration (cm/sec^2^)					
4th ventricle	5.68–12.5 (10.2)	Pre-Op	2.8–4.7 (3.4) *	2.1–3.2 (2.5) *	5.6–3.7 (2.8) *
		Post-Op	4.4–12.6 (11.3)	5.4–14.7 (12.8)	5.4–13.7 (10.9)
	Deteriorated cases: post-op 6 months		2.8–4.7 (3.4) *	2.1–3.7 (2.5) *	5.6–3.7 (2.8) *
Outlet of foramen of Magendie	5.70–12.6 (11.4)	Pre-Op	3.4 –7.5 (5.3) *	3.8–8.7 (4.4) *	3.3–7.2 (5.4) *
		Post-Op	5.9–14. (12.2)	6.8–18.7 (13.5)	5.9–7.5 (125)
	Deteriorated cases: post-op 6 months		3.3 –7.3 (5.0) *	3.5–8.7 (4.3) *	3.4–7.3 (5.4) *
Syrinx (spinal cord)	1.25–4.50 (2.80)	Pre-Op	1.7–4.7 (4.3)	1.8–4.2 (4.3)	1.0–4.8 (4.4)
		Post-Op	N.A.	N.A.	N.A.
	Deteriorated cases: post-op 6 months		1.45–3.50 (2.3)	1.28–32.40 (21.32)	1.44–32.33 (31.48)

*: significantly higher or more than other groups (*p* < 0.01). **: significantly lower or less than other groups (*p* < 0.01). Abbreviations: CM-I = Chiari malformation type I, Syrinx = syringomyelia, CSF = cerebrospinal fluid flow, N.A. = not applicable.

## Data Availability

All data are available on request to the corresponding author.

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
