# Peer review of "Surgical Management of Chiari Malformation Type I Associated with Syringomyelia: Outcome of Surgeries Based on the New Classification and Study of Cerebrospinal Fluid Dynamics"

_jcm, 2022, doi:10.3390/jcm11154556_

Round 1

Reviewer 1 Report

This study relates to the surgical treatment of patients with Chiari I malformation based on a relatively new classification of the causes of the hindbrain hernia.  The patients in the study had a herniation of the cerebellar tonsils at least 5 mm below the foramen magnum and carried the diagnosis of syringomyelia.  Multiple anatomic measurements of the posterior fossa and the brain itself were performed.  The authors had previously published the classification and how the measurements were performed.  Using this classification decisions were made related to what the goals of surgery were to be and how should the goals be attained.  The procedure and outcome of 330 patients are discussed.  There are slight differences in the numbers from the abstract and tables one and 2.  

It appears that this study relates to a subset of patients previously presented previously (authors reference 19). While presented in Table 1 the classification is difficult to understand.  It appears that Type A is a situation in which the tonsils are down and there are not abnormal measurements of the posterior fossa.  I assume but am not certain that this group includes those who ended up having a fusion either O-C1 or C1-C2 and have some abnormal movement as a cause of the distortion.  Type B relates to the standard CM-1 with need for increasing the volume of the posterior fossa.  There is little to know from the text whether there is always a duraplasty in the management of this condition.  Type C relates to a very small back of the skull such as that seen in Craniofacial abnormalities and requiring an expansion of the posterior calvarium although I cannot be certain of that.  The “borderline” cases are not further discussed but underwent posterior fossa decompression. 

The illustrative cases are well done and show what is being measured. Figures 4-2 and 4-1 should be shifted.

The authors show the measurements in the figures.  For a classification to be of value it needs to be usable for the purpose of prospective studies.  For that there needs to be more specific methodology for the measurements and the inter rater reliability.  With 8 authors it would seem that this would be easily accomplished and would make the study stronger.  

Author Response

Manuscript ID: jcm-1829882

Revision list for Reviewers

Surgical Management of Chiari Malformation Type I Associated with Syringomyelia: Outcome of Surgeries Based on the new Classification and Study of Cerebrospinal Fluid Dynamics

 Misao Nishikawa1,2.*, Toru Yamagata1, Kentarou Naito2,

Noritsugu Kunihiro3, Hiroaki Sakamoto3, Mistuhiro Hara1,2,

Kenji Ohata2,4, Takeo Goto2.

 1   Department of Neurosurgery, Moriguchi-Ikuno Memorial Hospital, Osaka Japan  

2   Department of Neurosurgery, Osaka Metropolitan University Graduate School of

   Medicine, Osaka Japan

3   Department of Pediatric Neurosurgery, Osaka City General Hospital,Osaka Japan

4   Department of Neurosurgery, Naniwa-Ikuno Hospital, Osaka Japan

*   Corresponding author : [email protected] TEL.: +81-90-7759-4035S

We deeply sincerely appreciate your review of our manuscript. The manuscript has been by a native English speaker.

We have revised the manuscript according to your suggestions as below. Please see my responses below.

We have highlighted the newly added and revised text.

Certification

The attached paper has been carefully reviewed by an experienced medical editor whose first language is English and who specializes in editing papers written by physicians and scientists whose native language is not English.

For Reviewer 1:

1: In accordance with the suggestion by Reviewer 2, I have revised the number in the Abstract and Table 1, excluding the cases with tethered cord syndrome, hydrocephalus, and other causes.

                The numbers are presented for each surgery, so the patients were included in multiple categories (i.e. FMD, ESCO, and CCF). We have confirmed the number of each operation.

2: We have revised the Background and Introduction, and highlighted the revised text.

  1. Background and Introduction

The mainstay of treatment of syringomyelia associated with Chiari malformation type I (CM-I) is the treatment of CM-I to normalize the cerebrospinal fluid (CSF) flow at the foramen magnum. Previously, we reported that CM-I patients have an underdeveloped occipital bone, leading to a shallow posterior cranial fossa (PCF) [1–3]. The shallow PCF results in the sagging of brainstem and cerebellum (hindbrain) into the spinal canal; therefore, the pathogenesis of CM-I is insufficiency of para-axial mesodermal, which is the origin of the occipital bone [1–10]. Therefore, foramen magnum decompression (FMD) to enlarge the foramen magnum is commonly performed for CM-I worldwide. However, other mechanisms of ptosis of the brainstem and cerebellum, including hypermobility and instability of the craniocervical junction (CCJ), traction by tethering, and pressure coning [3,11–15], may not respond to FMD. A lack of understanding of the mechanism of ptosis of the brainstem and cerebellum (hindbrain) in CM-I leads to inappropriate selection of the surgical treatment, which results in the recurrence of neurological symptoms and syringomyelia.

Since 2001, we have performed volumetric and morphometric studies of the PCF using magnetic resonance imaging (MRI) and computed tomography (CT) reconstructed images, which we previously established and evaluated for CM-I patients [1–3,11,12]. In addition, we also previously examined the mechanism of ptosis of the brainstem and cerebellum. On the basis of the mechanism of hindbrain ptosis, defined on the basis of volumetric and morphometric analyses, we classified CM-I into three independent subgroups: CM-I types A–C (Table 1; Fig.1) [16]. CM-I type B and CM-borderline are characterized by a normal volume of PCF (VPCF), small volume of the area surrounding the foramen magnum (VAFM), and small occipital bone size (Fig. 1B and 1D). CM-I type C is characterized by a small PCFV, VAFM, and occipital bone size (Fig. 1C), as well as elongation of the brain stem and downward displacement of the whole hindbrain. In CM-I types B and C, and CM- borderline, VBPCF/VPCF was large, i.e., the PCF was crowded.

In CM-I type A, anatomical abnormalities of the PCF (occipital bone size and hindbrain) were not found, rather only the downward displacement of the hindbrain was observed. CM-I type A (other pathogenesis) is caused by conditions such as craniocervical instability, tethered cord syndrome, hydrocephalus, intracranial mass lesions, and pressure dissociation (Tables 1 and 2; Fig. 1A, E–G) [16]. In this study, we evaluated CCJ hypermobility and instability as other mechanism of downward displacement of hindbrain.

We selected the surgical procedures for CM-I patients according to the mechanism of hindbrain ptosis (Table 2) [16]. For CM-I type A cases with CCJ hypermobility and instability, craniocervical fixation (CCF) was performed. For CM-I types B and C and CM-borderline, posterior fossa decompression (FMD or ESCP) was performed to correct PCF crowdedness. We present the preliminary results of treatments for the hindbrain ptosis in cases with CM-I associated with syringomyelia. We also examined the CSF flow dynamics at the foramen magnum pre- and post-surgery using cine phase contrast (Cine PC) magnetic resonance imaging (MRI). During surgery, CSF flow dynamics were evaluated using color Doppler ultrasonography (CDU) [17]. We assessed the target value to normalize the CSF flow at the foramen magnum.

Table 1.             Subtypes of Chiari malformation type I (CM-I) classified on the basis of morphometric analyses, cause of hindbrain ptosis, and surgical intervention.

Table 2.             Morphometric characterization and other mechanisms of hindbrain ptosis in CM- I type A, and the corresponding surgical intervention.

Figure 1. Illustrative cases: CM-I subtypes.

A: CM-I type A, B: CM-I type B, C: CM-I type C, and D: CM-borderline.

Left panel: 2D CT axial image at the level of the foramen magnum; right panel: magnetic resonance (MR) sagittal midline image.

E and F: MR midsagittal images demonstrating the character of CM-I type with HDCT. VPCF, VAFM, and occipital bone size are normal in size. E: MR midsagittal image in supine position showing normal interval BDI (7.7 mm), normal BAI (3.5 mm), normal CXA) (141°), large retro-odontoid pannus (asterisk), and low-lying cerebellar tonsills. F: On assumption of the upright position, there is evidence of cranial settling (2.5 mm decrease in BDI), posterior gliding of occipital condyle, 4.3 mm increase in BAI, anterior flexion of the occipito-atlantal joint (decrease of 8° in CXA), and increased cerebellar ptosis with downward displacement cerebellar tonsills to C1 (white arrow). Note the significantly increased impaction of the foramen magnum anteriorly and posteriorly. Bilateral black dotted arrow = CXA, asterisk = retro-odontoid punus, white arrow = tonsillar herniation. G: MR midsagittal image demonstrating the character of CM-I type A with TCS. VPCF, VAFM, and occipital bone size are normal in size. MR image showing elongation and downward displacement of the brainstem and cerebellum, as well as large supracerebellar cistern (double asterisk).

Abbreviations: CM-I= Chiari malformation type I, BDI= the interval between basion and dens, BAI= the interval between basion and atlas, CXA= the clivo-axial angle, HDCT = hereditary disorders of connective tissue, TCS = tethered cord syndrome.

3: We have added a descryiption of CM-borderline to the Background and Introduction, Materials and Methods, Resultsm and Discussion. The revised text is highlighted.

In Background and Introducetion.

and CM- borderline, VBPCF/VPCF was large, i.e., the PCF was crowded.

In materials and methods

2.4.1. Foramen magnum decompression (FMD) for CM-I type B and CM- borderline (Fig. 2)

We performed FMD for CM-I type B and CM-borderline because of VAFM crowdedness due to the small occipital bone size in these cases. The FMD involved craniectomy of the surrounding area of the foramen magnum (2–3-cm square), C1 laminectomy, and dural plasty for decompression of the brainstem and cerebellum, and normalization of CSF flow at the foramen magnum (Fig. 2A–D) [20].

In Results

3.2. Outcome of posterior fossa decompression (FMD and ESCP)(Table 4)

In patients aged ≥ 12 years, the improvement rate of neurological symptoms and signs in FMD was 86.5%. The JOA score RR in FMD was 58.7%, while 94.0% of cases showed improvement or stabilization of neurological symptoms. In 8 cases (6.2%), the neurological symptoms deteriorated during follow-up. In 10 (7.5%) out of 133 cases with syringomyelia, the symptoms persisted. The improvement rate of neurological symptoms and signs in ESCP was 89.7%. The JOA score RR in ESCP was 60.2%, and 98.9% of cases had an improvement or stabilization of neurological symptoms. In 2 cases (2.3%), the neurological symptoms deteriorated during follow-up. In 1 (1.1%) out of 87 cases with syringomyelia, the symptoms persisted. There were no significant differences between the FMD and ESCP groups in terms of improvement or stabilization of neurological symptoms and the JOA score RR. Persistent syringomyelia and deterioration of neurological symptoms and/or signs were significantly more common in patients who underwent FMD compared to ESCP (p > 0.01). In 8 patients aged 12 years or older with CM-I type B and CM- borderline who had deterioration of neurological symptoms and signs, ESCP was performed to stabilize the neurological symptoms and signs. In 2 patients with CM-I type C and deterioration of neurological symptoms and signs after ESCP, CCF was performed, which stabilized the neurological symptoms and signs.

In children aged younger than 12 years, improvement rate of neurological symptoms and/or signs were significantly lower in CM-I types A (58.3%) and C (64.3%) than those in CM-I type B (75.8%) and CM- borderline (71.4%) (p > 0.01). Furthermore, JOA score RR was significantly lower in CM-I type C (48.6%) than those in CM-I types A (57.2%) and B (58.2%), and CM- borderline (54.3%) (p > 0.01). Persistent syringomyelia and deterioration of neurological symptoms and/or signs were significantly more common in patients with CM-I types A (50.0% and 41.7%, respectively) and C (17.9% and 32.1%, respectively) than those in patients with CM-I type B (6.1% and 7.3%, respectively) and CM-borderline (0% and 14.3%, respectively) (p> 0.01).

In 6 children aged less than 12 years with CM-I type A who had deterioration of neurological symptoms/signs and/or CCJ instability, CCF was performed after 12 years of age, which resolved the neurological symptoms and signs. In 13 children aged less than 12 years and CM-I types B and C and CM-borderline who had deterioration of neurological symptoms/signs, ESCP was performed after 12 years of age, which resolved the neurological symptoms and signs.

In Discussion

4.2.          Outcome of surgical intervention

4.2.1. Outcomes of posterior fossa decompression (FMD and ESCP)

In patients younger than 12 years who underwent FMD for CM-I types B and C and CM-borderline, 13 cases had deterioration of neurological symptoms/signs and did not achieve adequate CSF space in the major cistern and normal CSF flow dynamics. Therefore, ESCP was performed after 12 years of age, which resolved their neurological symptoms/signs. In 8 patients aged 12 years or older who underwent FMD for CM-I type B and CM-borderline, there was deterioration of neurological symptoms/signs and adequate CSF space at the major cistern and normal CSF flow dynamics were not achieved. Therefore, ESCP was performed, which resolved their neurological symptoms/signs. Therefore, for CM-I type C patients and those who did not achieve adequate CSF space at the major cistern and normal CSF flow dynamics by FMD, the addition of ESCP was appropriate. The addition of ESCP should be considered for CM-I type C patients aged less than 12 years. In 10 preliminary cases with CM-I type C, we performed FMD and dural plasty initially, but could not achieve adequate CSF space volume and CSF maximum velocity; therefore, ESCP was performed. ESCP is a fundamental treatment for achieving adequate decompression of the hindbrain and normalization of CSF flow dynamics in patients with CM-I type C.

4: We shifted Fig. 4-2 and 4-1.

5: We have added a description regarding the methodology to the Materials and Methpds, as well ass Author contributeons.

added the description about methodology in Methods and Author contributions.

In Materials and Methods

2.3.2. Volumetric calculations, morphometric measurements and data collection.

Volumetric calculations and morphometric measurements were performed by three examiners who did not know the patients (M.N, T.Y, K.N and N.K.), excepting the main surgeon of the patient. Each three examiner calculated and measured three times and adopted the mead data.

In author contributions

and methodology; M.N, T.Y, K.N, and N.K; data curation; M.N, T.Y, and H.S;

6: We attempted to identify specific and simple measurement tools but were unable to identify any such tool. However, we will continues to explore such tools.

Reviewer 2 Report

 The manuscript is adequatley written and content is interesting for readers. The statistics and methods are good.

However, there are aspects that should be discussed and edited before the manuscript might be considered for publication:

1 review of English by a native speaker

2 the introduction is too short, please add more information about this topic

3 study design should be revised by excluding patients with associated conditions such as tethered cord syndrome and hydrocephalus (they may be causes of chiari I)

4 it is not clear why expansive suboccipital cranioplasty is better than FMD for CM-I type C, illustrate it in more details.

Author Response

Manuscript ID: jcm-1829882

Revision list for Reviewers

Surgical Management of Chiari Malformation Type I Associated with Syringomyelia: Outcome of Surgeries Based on the new Classification and Study of Cerebrospinal Fluid Dynamics

 Misao Nishikawa1,2.*, Toru Yamagata1, Kentarou Naito2,

Noritsugu Kunihiro3, Hiroaki Sakamoto3, Mistuhiro Hara1,2,

Kenji Ohata2,4, Takeo Goto2.

 1   Department of Neurosurgery, Moriguchi-Ikuno Memorial Hospital, Osaka Japan  

2   Department of Neurosurgery, Osaka Metropolitan University Graduate School of

   Medicine, Osaka Japan

3   Department of Pediatric Neurosurgery, Osaka City General Hospital,Osaka Japan

4   Department of Neurosurgery, Naniwa-Ikuno Hospital, Osaka Japan

*   Corresponding author : [email protected] TEL.: +81-90-7759-4035S

We deeply sincerely appreciate your review of our manuscript. The manuscript has been by a native English speaker.

We have revised the manuscript according to your suggestions as below. Please see my responses below.

We have highlighted the newly added and revised text.

Certification

The attached paper has been carefully reviewed by an experienced medical editor whose first language is English and who specializes in editing papers written by physicians and scientists whose native language is not English.

For Reviewer 2:

1: The manuscript has been revised by a native English speaker. We have attached a certificate of the review by the native English apeaker.

2: I have added this information to the Background and \Introduction because of the word limit (200 words) of the abstract. Therefor, the Background and Introduction has been revised, and the revise text is highlighted.

  1. Backgrounds and Introduction

The mainstay of treatment of syringomyelia associated with Chiari malformation type I (CM-I) is the treatment of CM-I to normalize the cerebrospinal fluid (CSF) flow at the foramen magnum. Previously, we reported that CM-I patients have an underdeveloped occipital bone, leading to a shallow posterior cranial fossa (PCF) [1–3]. The shallow PCF results in the sagging of brainstem and cerebellum (hindbrain) into the spinal canal; therefore, the pathogenesis of CM-I is insufficiency of para-axial mesodermal, which is the origin of the occipital bone [1–10]. Therefore, foramen magnum decompression (FMD) to enlarge the foramen magnum is commonly performed for CM-I worldwide. However, other mechanisms of ptosis of the brainstem and cerebellum, including hypermobility and instability of the craniocervical junction (CCJ), traction by tethering, and pressure coning [3,11–15], may not respond to FMD. A lack of understanding of the mechanism of ptosis of the brainstem and cerebellum (hindbrain) in CM-I leads to inappropriate selection of the surgical treatment, which results in the recurrence of neurological symptoms and syringomyelia.

Since 2001, we have performed volumetric and morphometric studies of the PCF using magnetic resonance imaging (MRI) and computed tomography (CT) reconstructed images, which we previously established and evaluated for CM-I patients [1–3,11,12]. In addition, we also previously examined the mechanism of ptosis of the brainstem and cerebellum. On the basis of the mechanism of hindbrain ptosis, defined on the basis of volumetric and morphometric analyses, we classified CM-I into three independent subgroups: CM-I types A–C (Table 1; Fig.1) [16]. CM-I type B and CM-borderline are characterized by a normal volume of PCF (VPCF), small volume of the area surrounding the foramen magnum (VAFM), and small occipital bone size (Fig. 1B and 1D). CM-I type C is characterized by a small PCFV, VAFM, and occipital bone size (Fig. 1C). In addition in CM-I type C, the elongation of the brain stem and downward displacement of the whole hindbrain. In CM-I type B, C, and CM- borderline, VBPCF/VPCF was large, this means the crowdedness in the posterior cranial fossa.

In CM-I type A, anatomical abnormalities in the posterior cranial fossa (occipital bone size and hindbrain) were not found, just downward displacement of the hindbrain was shown. CM-I type A (other pathogenesis) is characterized by conditions such as craniocervical instability, tethered cord syndrome, hydrocephalus, intracranial mass lesion and pressure dissociation (Tables 1 and 2; Fig. 1A, E, F, and G) [16]. In this study as the other mechanism of this downward displacement of the hindbrain the authors picked up the hypermobility and instability at CCJ.

We selected the surgical procedures for CM-I patients according to the mechanism of hindbrain ptosis (Table 2) [16]. For CM-I type A cases with hypermobility and instability at CCJ, craniocervical fixation was performed. For CM-I type B, C and CM-borderline, the posterior fossa decompression (FMD or ESCP) was performed to resolute this crowdedness of the posterior cranial fossa. We present the preliminary results of treatments for the hindbrain ptosis in cases with CM-I associated with syringomyelia. We also examined the CSF flow dynamics at the foramen magnum pre- and post-surgery using cine phase contrast (Cine PC) magnetic resonance imaging (MRI). During surgery, CSF flow dynamics were evaluated using color Doppler ultrasonography (CDU) [17]. We assessed the target value to normalize the CSF flow at the foramen magnum.

Table 1.             Subtypes of Chiari malformation type I (CM-I) classified on the basis of morphometric analyses, cause of hindbrain ptosis, and surgical intervention.

Table 2.             Morphometric characterization and other mechanisms of hindbrain ptosis in CM- I type A, and the corresponding surgical intervention.

Figure 1. Illustrative cases: CM-I subtypes.

A: CM-I type A, B: CM-I type B, C: CM-I type C, and D: CM-borderline.

Left panel: 2D CT axial image at the level of the foramen magnum; right panel: magnetic resonance (MR) sagittal midline image.

E and F: MR midsagittal images demonstrating the character of CM-I type with HDCT. VPCF, VAFM, and occipital bone size are normal in size. E: MR midsagittal image in supine position showing normal interval BDI (7.7 mm), normal BAI (3.5 mm), normal CXA) (141°), large retro-odontoid pannus (asterisk), and low-lying cerebellar tonsills. F: On assumption of the upright position, there is evidence of cranial settling (2.5 mm decrease in BDI), posterior gliding of occipital condyle, 4.3 mm increase in BAI, anterior flexion of the occipito-atlantal joint (decrease of 8° in CXA), and increased cerebellar ptosis with downward displacement cerebellar tonsills to C1 (white arrow). Note the significantly increased impaction of the foramen magnum anteriorly and posteriorly. Bilateral black dotted arrow = CXA, asterisk = retro-odontoid punus, white arrow = tonsillar herniation. G: MR midsagittal image demonstrating the character of CM-I type A with TCS. VPCF, VAFM, and occipital bone size are normal in size. MR image showing elongation and downward displacement of the brainstem and cerebellum, as well as large supracerebellar cistern (double asterisk).

Abbreviations: CM-I= Chiari malformation type I, BDI= the interval between basion and dens, BAI= the interval between basion and atlas, CXA= the clivo-axial angle, HDCT = hereditary disorders of connective tissue, TCS = tethered cord syndrome.

3: We have revised the data in Table 2 and 3, excludeing the cases with tethered cord, hydrocephalus, and iother causes, whicvh were descryibed that in the Exclusion criteria.

revised the data (Table 2, 3), excluding the cases with tethered cord, hydrocephalus and others and I described that in “2.2.2. Exclusion criteria”.

The cases with tethered cord syndrome hydrocephalus, pressure coning and pressure dissociation between the intracranial cavity and spinal canal were excluded, because these condition cause the ptosis of the hindbrain.

4: Se have added an explanation regarding why ESCP was appropriate for CM-I type C to the Discussion.

In discussion

4.2.               Outcome of surgical intervention

4.2.1. Outcomes of posterior fossa decompression (FMD and ESCP)

             In cases aged younger than 12 years, in CM-I types A and C, the improvement rate of neurological symptoms and/or signs was significantly lower than that in CM-I type B and CM-borderline. The rates of persistent syringomyelia and deterioration of neurological symptoms and/or signs were significantly much higher than those in CM-I type B and CM-borderline. These data suggest that in CM-I type C, FMD was not adequate to decompress the hindbrain or normalize the CSF flow at the foramen magnum; therefore, additional decompression should be considered. In children aged less than 12 years, for cases of CM-I type C, ESCP should be considered or two-staged surgery should be performed, which involves initial FMD followed by observation and ESCP.

In patients aged ≥ 12 years, ESCP and FMD were associated with good surgical outcomes. The outcomes in the present study were better than those reported previously in cases who underwent FMD only. In addition, morbidity and complications occurred less frequently in the present study than in previous studies [31–33]. These findings suggest that the selection of surgical procedures (i.e., ESCP or FMD) according to the morphometric analyses was appropriate. Similar to previous reports, the improvement rate for hindbrain ptosis after FMD and ESCP was high; however, a large proportion of patients with spinal cord myelopathy do not improve after surgery.

In patients younger than 12 years old, out of the cases who had FMD in CM-I type B, C and CM- borderline, 13 cases who had deterioration of neurological symptoms/signs did not achieved enough CSF space in the major cistern and normal CSF flow dynamics, so that ESCP was performed for them after 12 years old and their neurological symptoms/signs resolved. In aged 12 years old and older than 12 years, out of the cases who had FMD in CM-I type B and CM- borderline, 8 cases who had deterioration of neurological symptoms/signs did not achieved enough CSF space at the major cistern and normal CSF flow dynamics, so that ESCP was performed for them and their neurological symptoms/signs resolved. So that for CM-I type C cases or cases who did not achieved enough CSF space at the major cistern and normal CSF flow dynamics by FMD, the addition of ESCP was appropriate surgical procedure. We should consider the addition of ESCP for CM-I type C patients in child aged younger than 12 years old In CM-I type C, in preliminary 10 cases we performed FMD and dural plasty at first, but in this point the volume of CSF space and CSF maximum velocity did not achieve the goal, so we added ESCP. We think that ESCP is fundamental treatment which achieve enough decompression for the hindbrain, and enough CSF space and normalization of CSF flow dynamics at the foramen magnum for CM-I type C.

Round 2

Reviewer 1 Report

Very interesting study with interesting outcomes.  The problem overall is how much energy is needed to do the large number of data points are needed to come to the conclusions.